# A New Idea for RSA Backdoors

**Marco Cesati**

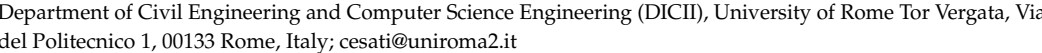

Department of Civil Engineering and Computer Science Engineering (DICII), University of Rome Tor Vergata, Via del Politecnico 1, 00133 Rome, Italy; cesati@uniroma2.it

**Abstract:** This article proposes a new method to inject backdoors in RSA (the public-key cryptosystem invented by Rivest, Shamir, and Adleman) and other cryptographic primitives based on the integer factorization problem for balanced semi-primes. The method relies on mathematical congruences among the factors of the semi-primes based on a large prime number, which acts as a "designer key" or "escrow key". In particular, two different backdoors are proposed, one targeting a single semi-prime and the other one a pair of semi-primes. This article also describes the results of tests performed on a SageMath implementation of the backdoors.

**Keywords:** RSA; backdoor; escrow key; implicit factorization problem; integer factorization

## 1. Introduction

Impairing the robustness of cryptographic applications is a sensitive topic. The interest in direct attacks, vulnerabilities, and backdoors for all currently used ciphers is certainly justified by economic and geopolitical reasons. If a vulnerable implementation of a cryptographic algorithm is surreptitiously distributed, an "evil" actor or a national security agency might easily access any sort of sensitive and precious information. However, "legal" actors might exist that openly mandate or encourage the adoption of cryptographic implementations that include backdoors, in order to realize "key escrow" mechanisms. For instance, a national country might legislate that judiciary representatives must always be able to recover any kind of encrypted communication involved in a criminal case.

Until a few years ago, it was only conjectured [1] that major security agencies were able to decrypt a large portion of the world's encrypted traffic, mainly thanks to vulnerabilities hidden in pseudo-random generators or major cryptographic algorithms and applications. Some examples of this practice might be the Hans Bühler case in 1994 [2], the Dual-EC algorithm proposed in 2004 by the US National Institute of Standards and Technologies [3,4], and perhaps the OpenBSD backdoor incident that emerged in 2010 [5,6]. However, in the last few years many government bodies have openly talked about enforcing by law "responsible encryption" or "exceptional access to encrypted documents" [7,8]: essentially, these are just more palatable words for "escrow key" and "backdoors".

The approach to backdoor construction has changed in the last few years. In the past, the focus was mainly on weaknesses in pseudo-random generators or software implementations that might allow attackers to predict some secret data of the target users. Nowadays, the emphasis is on theoretical backdoors based on mathematical properties of the cryptographic primitives. The main advantage of this new approach is that it is very difficult to discover a mathematical backdoor by just looking at the cryptographic algorithm. For example, Bannier and Filiol [9] showed in 2017 how a block cipher similar to the Advanced Encryption Standard (AES) can be devised so that it includes, by design, a hidden mathematical backdoor that allows a knowledged attacker to effectively break the cipher and recover the key.

Evil actors and legal actors pursue very different goals. This fact justifies the adoption of very different backdoor mechanisms. An evil actor is primarily concerned with how convenient triggering the backdoor is and secondarily how well the backdoor mechanism is

hidden from the final user; however, it is not crucially important to also preserve the security of the cipher. Thus, a backdoor introduced by an evil actor might even be a vulnerability hidden in a cipher implementation such that anyone knowing about its existence could easily break the cipher and recover the encrypted messages. For instance, a mechanism that can be easily exploited might be based on a semi-prime generator that selects just one of the primes at random, while the other prime is fixed. The Euclidean algorithm applied to two different vulnerable semi-primes outputs the fixed prime; thus, anyone can easily break the cipher even if the fixed prime is not known in advance. Perhaps not surprisingly, there are a lot of very weak public keys in the Internet [10,11]. However, a legal actor does not want to significantly impair the security of a cryptographic algorithm, because the final users might just refuse to adopt an insecure cipher. A backdoor introduced by legal actors is likely a vulnerability embedded in a cryptographic implementation that allows only "authorized" actors to decipher the encrypted messages without knowing the private keys of the final users. Usually, this means that the retrieval of the encrypted messages can be performed only if the actor knows a secret escrow key related to the backdoor itself.

Among the most widespread cryptographic algorithms, RSA [12] deserves special consideration, because it is conveniently used to protect any kind of sensitive data transmitted over the Internet. It is commonly believed that RSA has been properly designed and that, by itself, it does not contain hidden vulnerabilities. However, a large number of attacks to RSA have been proposed since its invention. These attacks span from directly factoring the semi-prime in the public key to exploiting weaknesses in the generation algorithm for the prime factors; for a survey, see [13]. Furthermore, several RSA backdoors have been proposed: they are specially crafted values in RSA parameters that allow a knowledgeable attacker to recover the private key from publicly available information. For an in-depth discussion of several RSA backdoors, see [14].

This article proposes a new idea to inject backdoors in RSA key generators, which was loosely inspired by the concept of "implicit hints" of May and Ritzenhofen [15] in pairs of semi-primes. However, this idea differs significantly from the backdoors based on implicit hints and, as far as I know, from any other published backdoor proposal (a preliminary version of this work has been published as a preprint [16]).

More specifically, May and Ritzenhofen proposed the implicit factorization problem (IFP), which is based on the premise that two or more semi-primes with factors sharing some common bits can be factored with some variants of the Coppersmith's algorithm [17,18]. The authors stated that "[. . . ] one application of their result is malicious key generation of RSA moduli, i.e., the construction of backdoored RSA moduli". In my opinion, however, a backdoor based on shared bits, as described in [15], is not really effective for RSA. It is practically not possible to exploit this backdoor in large "balanced" semi-primes, such as those used in currently used RSA moduli, because the time required by the Coppersmith's algorithm to factor a semi-prime grows exponentially as the size difference of the factors becomes smaller. Moreover, this vulnerability is self-evident to anyone looking at the factors, because there would be a long run of identical bits in the two values; thus, such backdoors cannot be easily concealed from the owner of the private keys.

The new idea is the following: rather than including in the bit expansions of the factors a long run of *identical* bits, the bit expansions include portions of *correlated* bits, where the correlation is bound to a secret designer key not known to the owner of the backdoored keys. In practice, the backdoor designer enforces some mathematical conditions on the values of the factors, such as congruences with a modulo for a large prime (of nearly the same size of the factors), which acts as the designer key.

Following the IFP approach in [15], I firstly devised a backdoor (named the TSB, Twin Semi-prime Backdoor) based on mutual correlations between the factors of two distinct semi-primes. Afterwards, I devised a simpler backdoor (named the SSB, Single Semi-prime Backdoor) based on the same idea but suitable for injecting a backdoor in a single semi-prime. The backdoors can be applied to RSA and also to any other cipher whose security is based on the difficulty of the integer factorization of semi-primes.

A key difference from the IFP approach is that in triggering the backdoors, that is, in order to factor the semi-prime(s) by exploiting the designer key, there is no need to apply some variant of the Coppersmith's algorithm. Therefore, if the value of the designer key is known, factoring the semi-prime(s) is easy and efficient. However, if the designer key is not known, there seems to be no efficient way to factor the semi-prime(s). Moreover, without the designer key, there seems to be no efficient way to detect the existence of the backdoor, even when looking at the distinct prime factors of the semi-prime(s). Of course, significant progresses in quantum computing might affect the robustness of the proposed backdoors. However, such progresses will likely have a significant impact on all aspects of the RSA algorithm.

The rest of the article is organized as follows. Section 2 includes some mathematical notations and an introduction to the basic RSA algorithm. Section 3 includes a discussion of the prior works related to RSA backdoors and the implicit factorization problem (IFP). Section 4 presents the simpler backdoor, the SSB, while Section 5 presents the more sophisticated backdoor for a pair of semi-primes, the TSB. Finally, Section 6 includes the conclusions of this work.

## 2. Preliminaries

In this article, $a \equiv b \pmod{c}$ denotes the relation in which $a - b$ is a multiple of $c$; often the shorter notation $a \equiv_c b$ will be used. The notation $a \bmod b$ denotes the operation remainder of the division $a/b$; hence, $a \equiv_c (a \bmod c)$ and $0 \leq a \bmod c < c$.

If $N \geq 0$ is an integer, its size in bits is defined as $\ell(N) = \max\{1, \lceil \log_2(N + 1) \rceil\}$. Writing $x \simeq y$ means that $x$ and $y$ are equal or differ by at most one, while $x \approx y$ means that $x$ and $y$ differs by a value negligible with respect to the sizes of $x$ and $y$. If $N \approx 2^n$, for any large $n$, then $\ell(N) \simeq \log_2 N$; that is, both $\ell(N)$ and $\log_2 N$ may be considered to be approximately equal to $n$, ignoring a $\pm 1$ difference in size.

If $h$ is an integer, $h\rceil^k$ denotes the $k$ most significant bits of $h$ (a value from 0 to $2^k - 1$), while $h\rfloor_k$ denotes the $k$ less significant bits.

A *semi-prime* is a number $N$ such that $N = pq$ where $p$ and $q$ are primes. Therefore, $\ell(N) \simeq \ell(p) + \ell(q)$. If $\ell(p) \simeq \ell(q)$, then the semi-prime is said to be *balanced*. When considering sequences of semi-primes $N_i = p_i q_i$ ($i = 1, 2, \ldots$), it is assumed that they have a common size $n = \ell(N_i)$, for every $i$; furthermore, the primes $q_i$ have a common size $\ell(q_i) = \alpha$; it follows that all primes $p_i$ have the same size $n - \alpha$.

The RSA public key cryptosystem was invented by Rivest, Shamir, and Adleman [12] in 1977. In its simplest form, the algorithm is based on a balanced semi-prime $N = pq$ and a couple of exponents $e, d$ such that $\gcd(e, \phi(N)) = 1$ and $ed \equiv 1 \pmod{\phi(N)}$. Here, $\phi(N)$ denotes the Euler's totient function, which can be easily computed as $(p - 1)(q - 1)$ if the prime factors $p$ and $q$ are known. Theoretically, the value of $e$ could be random, while the value of $d$ can be computed from $e$ and $\phi(N)$ using the Extended Euclidean algorithm. The pair $(N, e)$ is the "public key" of RSA, and the encryption function is $M^e \bmod N$. Either the pair $(p, q)$ or the pair $(N, d)$ is the "private key", and the decryption function is $(M^e)^d \bmod N \equiv M^{s\phi(N)+1} \equiv M(M^{s\phi(N)}) \equiv M(1^s) \equiv M \pmod{N}$. Of course, factoring $N$ allows an attacker to recover the private key from the public key, because from $p$ and $q$ we can compute $\phi(N)$ and then $d \equiv e^{-1} \pmod{\phi(N)}$.

## 3. Related Work

Many authors proposed to classify backdoors embedded in cryptographic applications according to several, different criteria. Following [19], there exist three types of backdoors: (1) weak backdoors, (2) information transfer via subliminal channels, and (3) SETUP mechanisms. Weak backdoors are based on modifications of the cryptographic protocol such that it would be possible for anyone to break the cipher and recover the secret data. Vulnerabilities falling under the information transfer via subliminal channels category allow an attacker to exploit the cryptographic protocol in such a way to create a hidden communication channel that cannot be intercepted or unambiguously detected. Finally, SETUP

(Secretly Embedded Trapdoor with Universal Protection) mechanisms create vulnerabilities in the cryptographic protocols that cannot be easily exploited by third-party attackers.

SETUP mechanisms were firstly proposed by Young and Yung [20,21] in 1996: they coined the term "kleptography" to denote the usage of cryptographic primitives in order to design "safe" backdoors in other cryptographic protocols. Following the classical distinction between asymmetric and symmetric cryptography, SETUP mechanisms can lead to *asymmetric backdoors* and *symmetric backdoors*.

In an asymmetric backdoor, the information required to recover the encrypted messages is protected by an asymmetric cipher. Usually, this means that some data that allow an actor to recover any user private key are encrypted with the public key of the designer of the RSA implementation and stored inside the corresponding user public key. Any actor that knows the corresponding designer private key may extract the data from the user public key and decipher them to recover the user private key. Notice that in this case the RSA implementation is "tamper resistant": even reverse engineering cannot reveal the designer private key.

In a symmetric backdoor, however, the designer key that allows an actor to recover the user private key from the user public key is stored in some form inside the RSA implementation itself. To be secure and undetected, the RSA implementation (perhaps a physical device) must be "tamper proof".

Existing RSA backdoors may also be categorized according to the place where the backdoor's specific data are stored: either in the semi-prime $N$ alone or also in the exponent $e$ of any public key $(N, e)$. "Exponent-based" backdoors are somewhat easier to devise, because $e$ could theoretically be any random value coprime with $\phi(N)$. However, most RSA implementations make use of special fixed values for the public exponent, such as small values or values with a small Hamming weight, in order to improve the efficiency of the RSA algorithm. Thus, exponent-based backdoors cannot be easily hidden from the final user and can be perceptively slower than honest RSA implementations. Backdoors embedded in the public key's semi-prime do not limit the choice of the public exponent; however, they must address a crucial problem: how to encode information about the factorization of the semi-prime in the semi-prime itself, in such a way that the information is encrypted with a secret key and, possibly, the pair $(p, q)$ is indistinguishable from a pair of primes generated by an honest RSA implementation.

This article proposes two backdoors embedded in the semi-primes of the RSA's public keys; as a matter of fact, the backdoors apply to any cryptographic protocol based on the integer factorization of semi-primes. Related work concerning exponent-based backdoors is not further discussed here; examples of these backdoors can be found in [14,22–24].

### 3.1. Symmetric Backdoors

The proposed SSB algorithm implements a symmetric backdoor, because the escrow key is fixed and hard-cabled in the hardware or software device that generates the vulnerable semi-primes. As we shall see, the TSB might be considered both a symmetric or an asymmetric backdoor.

The first RSA backdoor was proposed by Anderson [25] in 1993. It is a symmetric backdoor embedded in the public key's semi-prime: let $\beta$ be an $m$-bit secret prime (the "backdoor key"), and let $\pi_\beta$ and $\pi'_\beta$ be pseudo-random functions that, given a seed in argument, produce a $(n - m)$-bit value (in the original article, $n = 256$ and $m = 200$). For any vulnerable $2n$-bit semi-prime $N = pq$, let $t, t' < \sqrt{\beta}$ be $(m/2)$-bit random numbers that coprime with $\beta$, and let $p = \pi_\beta(t) \cdot \beta + t$ and $q = \pi'_\beta(t') \cdot \beta + t'$. Given $N$ and $\beta$, it is possible to compute $tt' = N \bmod \beta$, then factor the $m$-bit number $tt'$, and finally compute $p$ and $q$. Kaliski [26] proved that it is possible to discover the backdoor by either computing the continued fraction $p/q$, because the expansion likely contains an approximation of the fraction $\pi_\beta(t)/\pi'_\beta(t')$, or by finding a reduced basis of a suitable lattice built on the primes of two vulnerable moduli. He also showed that the backdoor can be detected by the lattice method when 14 or more non-factored vulnerable moduli are available. It is

easy to observe that Kaliski's detection algorithm can be easily defeated by introducing a "dynamic backdoor key" whose exact value depends, for instance, on an incremental counter. However, another drawback of Anderson's backdoor is that $m \approx 3/4 \cdot n$; hence, triggering the backdoor for currently used public key sizes might require factoring a too large integer.

The first backdoor proposed in this article, the SSB, is similar to Anderson's construction, in that triggering the backdoor involves as first step computing the remainder of the integer division of the semi-prime and the designer (escrow) key. However, a key difference from Anderson's idea is the form of the primes $p$ and $q$, which allows the SSB to escape detection by Kaliski's algorithms and to avoid factoring a large integer when exploiting the backdoor.

In 2003, Crepéau and Slakmon [23] presented, among several other exponent-based backdoors, a semi-prime-based backdoor that relies on Coppersmith's attack [18] and encrypts the factor $p$ in the RSA modulus $N = p\,q$ in such a way that the bits in $N \rceil^{n/8}$ have the correct distribution for a random semi-prime, while the middle $n/4$ bits of $N$ are an encryption, via a pseudo-random function $\pi_\beta$, of $p \rceil^{n/4}$. The SSB and TSB backdoors use an entirely different mechanism and do not rely on Coppersmith's attack, which means that they can be efficiently exploited even on very large balanced semi-primes.

In 2008, Joye [27] studied the performances of generating a semi-prime $N$ in which some bits are prescribed; he developed as an example an RSA symmetric backdoor based on the Coppersmith's attack in which some of the bits of $p$ are encrypted in $q$. While this study is relevant when analyzing the generation times of any semi-prime backdoor, their proposal is entirely different than the present one.

The symmetric backdoor proposed by Patsakis [28] in 2012 is based on yet another idea: the parameterized, randomized backdoor algorithm decomposes an integer as a sum of squares in a way depending on a designer's secret parameter. The backdoor consists of imposing that the semi-prime, once decomposed using the secret parameter, can be easily solved by a nonlinear system whose solutions are properly bounded.

In 2017, Nemec, Sys, and others [29] exposed ROCA (Return of Coppersmith's Attack), a critical vulnerability (perhaps unintentional) in the key generation algorithm of the *RSALib* library, which is written, adopted, and distributed to third parties by Infineon, one of the top producers of cryptographic hardware devices. This work raised much interest because the flaw was already present in devices produced in 2012 and the total number of affected devices and, consequently, vulnerable keys is huge. In any $N = p\,q$ generated by the flawed *RSALib*, all primes $p$ and $q$ have the form $k \cdot M_t + (65,537^a \bmod M_t)$, where $M_t$ is the *primorial* number composed by the product of the first $t$ primes, and $k$, $a$ are random integers. The values of $t$ for semi-primes of bit length $n = 512, 1024, 2048$, and $4096$ are, respectively, $t = 39, 71, 126$, and $225$. This means that the number of truly random bits in each of the primes is reduced, respectively, to $98, 171, 308$, and $519$. In order to find the factors of a vulnerable semi-prime, a variant of the Coppersmith's attack is used: it is possible to efficiently factor $N = p\,q$ when the value $p \bmod M$ is known. Hence, the recovering procedure determines a suitable divisor $M$ of $M_t$ of size $\ell(M) \geq n/4$ (to reduce the search space for $a$), guesses an exponent $a$, computes $67,537^a \bmod M$, and factors $N$. It is also easy to verify whether a given key is flawed: $N$ is likely vulnerable if the discrete logarithm $\log_{65,537} N \bmod M_t$ exists. Actually, this logarithm can be easily computed by the Pohlig–Hellman algorithm [30] because $M_t$ is the product of many small consecutive primes. Hence, ROCA arguably belongs to the weak backdoor category.

### 3.2. Asymmetric Backdoors

The proposed TSB algorithm can be used to implement both symmetric and asymmetric backdoors. In fact, the TSB makes use of an embedded designer key but also generates two distinct semi-primes. If both semi-primes are used to build two distinct public keys, both available to a third-party attacker, then tampering with the TSB device may expose the designer key and break the keys. The TSB can be used to generate a public key (from one of

the generated semi-primes) and a dedicated escrow key composed by the hard-coded large prime inside the device and the other semi-prime, which must be considered the designer's secret key. This is a reasonable scenario for cryptographic keys used in a highly-secure work environment. In this second case, the TSB must be considered an asymmetric backdoor, because tampering with the device is not enough to break an already generated key.

The first examples of asymmetric backdoors proposed by Young and Yung [20] in 1996 were exponent-based. However, that article also includes the description of an asymmetric semi-prime-based backdoor named PAP, for "Pretty Awful Privacy". The backdoor designer defines a designer's RSA public key $(N' = p'q', e')$ and private key $(p', q', d')$, where $\ell(N') = n/2$. Let $F_K$ and $G_K$ be invertible functions depending on a fixed key $K$ that transform a seed of $n/2$ bits in a pseudo-random value of $n/2$ bits. In order to create a backdoor, the designer first chooses a prime $p$ of bit length $n/2$ at random then searches the smallest value $K$ such that $\rho = F_K(p) < N'$. $\rho$ is then encrypted as $\rho_2 = G_K(\rho^{e'} \bmod N')$. The RSA semi-prime $N$ results from the search of a prime $q$ such that the $n/2$ most significant bits of $N = p\,q$ coincide with $\rho_2$. The attacker can easily break the public key by extracting $\rho_2$ from $N$ then starting an exhaustive search of the value for $K$ that, when applied to the inverse permutations $G_K^{-1}$ and $F_K^{-1}$, permits the extraction of the proper factor $p$ using the RSA private key $(p', q', d')$.

In a series of articles published between 1997 and 2008, Young and Yung [21,31–33] proposed several kleptographic backdoors for RSA using different cryptographic algorithms for embedding the factor $p$ in $N$. Specifically, in [21] the backdoor PAP2 is embedded in the RSA semi-prime via the ElGamal protocol [34]; that is, encrypting $p$ in $N$ is based on a Diffie–Hellman key exchange. In [31] the backdoor PP, for "Private Primes", is based on Rabin's cryptosystem [35]; it also differs from the one described in [21] because it uses non-volatile memory to store the number of generated backdoored keys so as lower the probability of producing the same key twice. In [32] the encryption of the factor $p$ inside the semi-prime $N$ is achieved by means of an elliptic curve Diffie–Hellman key exchange. In 2008, Young and Yung [33] revisited the backdoor proposed in [32] and implemented it on the OpenSSL library. After some optimization effort, this implementation was made faster than the original OpenSSL RSA key generation methods.

In 2010, Patsakis [28,36] proposed yet another kleptographic mechanism that relies on Coppersmith's attack and forges $p$ and $q$ so that the most significant bits of both of them are of the form $(a + r)^{e'} \bmod N'$, where $a$ is a secret design parameter, $r$ is a random value, and $(N', e')$ is the designer's asymmetric public key.

In 2016, Wüller, Kühnel, and Meyer [37] proposed an RSA backdoor called PHP, for "Prime Hiding Prime", in which the information required to factor $N$ is hidden in $N$ itself. The idea is to select a prime $p$ such that $q = (p^{e'} \cdot p^{-1}) \bmod N'$ is a prime, where $(N', e')$ is the RSA public key of the designer. To factor $N = p\,q$, the designer computes $N^{d'} \equiv_{N'} (p \cdot p^{e'} \cdot p^{-1})^{d'} \equiv_{N'} p$. An improvement of PHP, called PHP', is also described in [37]: here, $q = (s^{e'} \cdot p^{-1}) \bmod N'$, where $s$ is the concatenation of $n/4$ random bits and $p\rfloor_{n/4}$. Half of the bits of $p$ are enough to recover the factorization of $N$ thanks to the Coppersmith's attack.

Markelova [19] revisited Anderson's idea for a symmetrical backdoor and devised SETUP mechanisms that protect the backdoor by means of some public key algorithms, in particular, based on discrete logarithm problems on both finite fields and elliptic curves. The author also presented a SETUP backdoor exploiting the Chinese Remainder theorem. The article [19] also includes a discussion of the similarities between these SETUP backdoors and the ROCA backdoor.

### 3.3. The Implicit Factorization Problem

In 1985, Rivest and Shamir [38] introduced the *oracle complexity* as a new way to look at the complexity of the factorization problem (and the related RSA attack): they showed that the semi-prime $N$ can be factored in polynomial time if an oracle provides $3/5$ of the bits of $p$. In 1996, Coppersmith [17,18] improved the result by showing that

an explicit "hint" about the top half bits of $p$ are sufficient for factoring $N$ in polynomial time. In particular, Coppersmith described some algorithms based on lattice reduction and the LLL procedure [39] to find small integer roots of univariate modular polynomials or bivariate integer polynomials. Later [40,41], these algorithms were reformulated in simpler ways and heuristically extended to the multivariate polynomial case.

The seminal article [15] focusing on "implicit hints" was published in 2009 and it is due to May and Ritzenhofen. An oracle gives an implicit hint when it does not output the value of some bits of one of the factors of the semi-prime; rather, the oracle outputs another semi-prime whose primes share some bits with the factors of the original semi-prime. The authors formally introduced the implicit factorization problem (IFP) and showed that two semi-primes $N_1$ and $N_2$ can be factored in time $O(n^2)$ if $p_1\rfloor_t = p_2\rfloor_t$, with $t \geq 2\alpha + 3$. The algorithm is based on a lattice reduction: the search for the unknown primes $q_i$ is reduced to a search for a basis of a suitable lattice by means of the quadratic Gaussian reduction algorithm. This result implies that only highly imbalanced semi-primes can be factored, because $\ell(q_1) = \ell(q_2) = \alpha$; hence, $\ell(p_i) > 2\,\ell(q_i)$. The authors also extended this result to $k > 2$ semi-primes and showed that a polynomial algorithm based on the Lenstra-Lenstra-Lovász lattice basis reduction (LLL) algorithm [39] exists if $t \geq \alpha k/(k-1)$. For the balanced case, this result is not useful, because it means that all $p_i$ primes are identical; hence, they can be easily recovered by the Euclidean algorithm. However, the authors also showed that their method can be used to factor $k$ balanced semi-primes when some additional conditions are satisfied and $n/4$ bits are discovered by brute force.

In the following years, many articles improved and extended the results of May and Ritzenhofen: further details can be found in a survey [42] published in 2018.

All attacks and vulnerabilities based on these results assume that the factors of vulnerable semi-primes share some identical bits. From a practical point of view, backdoors relying on shared identical bits cannot be easily concealed from anyone looking at the factors, that is, from the owner of the private key. Furthermore, all the results cited in this section are based on some variants of Coppersmith's algorithms [17,18]. In contrast, the proposed backdoors generate semi-primes with factors without common shared bits and do not require Coppersmith's algorithm. Therefore, they are difficult to detect and are much more efficient when applied to balanced semi-primes with a large size, such as those used in the current RSA public keys.

## 4. SSB: A Backdoor Embedded in a Single Semi-Prime

This section presents the SSB (Single Semi-prime Backdoor), a proposal for a new backdoor encoded in the value of a semi-prime $N$. The section first describes the vulnerability and how the semi-prime is generated; then, it describes the procedure to efficiently factor the semi-prime, provided that the corresponding escrow key is known. Finally, the section reports an analysis of the theoretical and practical efficiency of the backdoor.

### 4.1. Generation of a Vulnerable Semi-Prime

The first step of the generation of a vulnerable semi-prime is choosing an "escrow key", which is a secret value that must be known in order to detect and exploit the backdoor. The escrow key is a prime $T$ of a size slightly smaller than the size of the primes in the semi-prime. Thus, if $\alpha$ is the reference bit length of the primes (e.g., $\alpha = 512$ for RSA-1024), then $\ell(T) \simeq \alpha - c$, where typically $4 \leq c \leq 10$ for $\alpha \leq 2048$; a good value for $\alpha = 512$, 1024, and 2048 appears to be $c = 7$. The backdoor designer must also choose the value of a constant $K$, which must be related to the value of $\alpha$, as discussed later; in particular, it will be shown that a good practical choice may be $K \approx \alpha$.

In order to create a vulnerable semi-prime, two distinct primes $p$ and $q$, each of them having a bit length of roughly $\alpha$, must be generated. The backdoor exists whenever the following condition holds:

**H0.** *There exists a positive integer $k$ with $1 < k \leq K$ such that $p \equiv_T k\,q$.*

Algorithm 1 below can be used to generate the two primes $p$ and $q$ satisfying the condition H0. It is based on Dirichlet's theorem stating that there are infinitely many primes of the form $a + b\,c$ if $\gcd(a, b) = 1$ (consider $b = T$ and $a = k \cdot (q \bmod T)$). The semi-prime is then computed as $N = p\,q$.

---

**Algorithm 1:** Generation of a vulnerable semi-prime with escrow key $T$

---

*GetCorrelPrimes***:**
    **Input** : $\alpha, T, K$
    **Output**: $p, k$

    **while** *true* **do**
        generate random primes $q$ and $r$ of size $\alpha$
        **for** $k \leftarrow 2$ **to** $K$ **do**
            $p \leftarrow r + (k\,q - r) \bmod T$
            **if** *p is prime* **then**
                **return** *p, q*
            **end**
        **end**
    **end**

---

*4.2. Recovering Procedure*

The key idea of the SSB, and also the proof that it works as expected, is its recovering procedure. Formally, the factors of $N$ can be efficiently recovered by knowing in advance only the semi-prime $N$ and the escrow key $T$. The values of the parameters $\alpha$, $K$, and $c$ may affect the running time of the recovering procedure; however there is no need to know them to recover the factors.

The recovering procedure can be split into three phases:

1. Recovering "low-level" coefficients.
2. Recovering "high-level" coefficients.
3. Recovering the factors.

Generally speaking, in a practical implementation of the recovering procedure it might be convenient to interleave the executions of these three phases. However, here the phases are discussed independently to simplify the description of the whole procedure.

**Example 1.** *A "running example" may be useful to understand the description of the SSB's recovering procedure. Let $\alpha = 128$, $c = 5$, $K = 30$. Pick as a random secret the 123-bit prime $T = 6451117418610792529759522664972769997$. Then, pick as vulnerable semi-prime $N = 5457768026042466571066314310612087465251911219452327782472161824579382995\,4991$ (of bit length 255).*

4.2.1. Recovering "Low-Level" Coefficients

At the beginning, only $N$ and $T$ are known. The equation $N = p\,q$ and the equation in condition H0 imply the following:

$$N \bmod T \equiv_T (p \bmod T) \cdot (q \bmod T) \tag{1}$$

$$p \bmod T \equiv_T (k\,q) \bmod T \tag{2}$$

By combining them, we obtain the following:

$$N \bmod T \equiv_T (k\,q^2) \bmod T \tag{3}$$

Because $k \in [2, K]$, where $K$ is a reasonably small constant, we can exhaustively test every possible value for $k$ and discard any value for which $N \cdot k^{-1}$ in the Galois field $GF(T)$ is a quadratic non-residue, that is, discard any value $k$ such that for all integers $\gamma \in [0, T)$, $(N \bmod T)\,(k^{-1} \bmod T) \not\equiv_T \gamma^2$. Here, $k^{-1}$ denotes the value in $GF(T)$ such that $k \cdot k^{-1} \equiv_T 1$.

The output of this phase is a list containing candidate values for the "low-level" coefficient $k$ and the corresponding quadratic residue $\gamma^2$ in $GF(T)$. The correct value of $k$ yields $\gamma^2 \equiv_T q^2$.

**Example 2** (Continuing Example 1). *There are 14 values for $k \in [2, 30]$ that yield a quadratic residue in GF(T). They are 3, 4, 9, 10, 12, 13, 14, 16, 19, 22, 23, 25, 27, and 30.*

4.2.2. Recovering "High-Level" Coefficients

This phase starts by knowing $N$, $T$, $k$, and $q^2 \bmod T$. Actually, this phase is executed once for any candidate in the list built in the previous phase; any candidate is discarded as soon as it yields inconsistent results.

The first step computes the square root of $\gamma^2 = q^2 \bmod T$ in $GF(T)$; that is, it finds the values whose square is congruent to $\gamma^2$ modulo $T$, typically by means of the Tonelli–Shanks algorithm [43,44]. Because in general any square root has two distinct values in $GF(T)$, there are two possible values $\gamma_1$ and $\gamma_2$ for $q \bmod T$, where $\gamma_1 \equiv_T T - \gamma_2$. In the following, let $\gamma$ be either $\gamma_1$ or $\gamma_2$; this phase has to be performed with both values by discarding the value that yields inconsistent results.

Starting from $q \bmod T$, the value $p \bmod T$ can be easily computed from Equation (2), so several candidate values for $q \bmod T$ and $p \bmod T$ are now known.

**Example 3** (Continuing Example 2). *The 14 possible values for $k$, each of them with two possible roots $\gamma_1$ and $\gamma_2$, yield the following 28 cases:*

| $k, \gamma$ | $q \bmod T$ | $p \bmod T$ |
|---|---|---|
| $3, \gamma_1$ | 1101001108223132047246029465205384188 | 3303003324669396141738088395616152564 |
| $3, \gamma_2$ | 5350116310387660482513493199767385809 | 3148114093941396388021434269356617433 |
| $4, \gamma_1$ | 383884601054424720447564657194317617 | 1535538404217698881790258628777270468 |
| $4, \gamma_2$ | 6067232817556367809311958007778452380 | 4915579014393093647969264036195499529 |
| $9, \gamma_1$ | 255923067369616480298376438129545078 | 2303307606326548322685387943165905702 |
| $9, \gamma_2$ | 6195194351241176049461146226843224919 | 4147809812284244207074134721806864295 |
| $10, \gamma_1$ | 674267825617802548964398838956350795 | 291560837567232959884465724590737953 |
| $10, \gamma_2$ | 57768495929929899980795123826016419202 | 6159556581043559569875056940382032044 |
| $12, \gamma_1$ | 550500554111566023623014732602692094 | 1548892302729997537166541262595355131 |
| $12, \gamma_2$ | 5900616864499226506136507932370077903 | 6296228187882792776042868538713234866 |
| $13, \gamma_1$ | 872807543698631712198073475805281438 | 4895380649471419728815432520495888697 |
| $13, \gamma_2$ | 55783098749121608175614491891671488559 | 1555736769139372800944090144476881300 |
| $14, \gamma_1$ | 1772631623417650051858813089627283653 | 5463490472014723136744815259863661151 |
| $14, \gamma_2$ | 4678485795193142477900709573454863484 | 987626946596069393014707405109108846 |
| $16, \gamma_1$ | 3033616408778183904655979003889226190 | 3380040610175394766179005407418229061 |
| $16, \gamma_2$ | 3417501009832608625103543661083543807 | 3071076808435397763580517257554540936 |
| $19, \gamma_1$ | 1334962546318133547479911059450973176 | 6010936124212159812839742134650180353 |
| $19, \gamma_2$ | 5116154872292658982279611605521796821 | 440181294398632716919780530322589644 |
| $22, \gamma_2$ | 392162883320122101182846126731882268 | 2176466014431893696263092123128639899 |
| $22, \gamma_2$ | 6058954535290670428576676538240887729 | 4274651404178898833496430541844130098 |
| $23, \gamma_1$ | 2533078726893509165415881053303209543 | 200753951053578036729560241218889516 |
| $23, \gamma_2$ | 3918038691717283364343641611669560454 | 6250363467557214493029962423753880481 |
| $25, \gamma_1$ | 2426893127022547123724783203111380952 | 2612271408066545325283876093029593827 |
| $25, \gamma_2$ | 4024224291588245406034739461861389049 | 3838846010544247204475646571943176170 |
| $27, \gamma_1$ | 367000369407710682415343155068461396 | 3457892555397395895454742521875687695 |
| $27, \gamma_2$ | 6084117049203081847344179509904308601 | 2993224863213396634304780143097082302 |
| $30, \gamma_1$ | 2107797709484122639489264125803469073 | 5173874517026546416842219789349142217 |
| $30, \gamma_2$ | 4343319709126669890270258539169300924 | 1277242901584246112917302875623627780 |

The semi-prime $N$ can be written as follows:

$$p\,q = (\pi\,T + (p \bmod T)) \cdot (\nu\,T + (q \bmod T)), \qquad (4)$$

that is, if $\delta = (N - (p \bmod T)(q \bmod T))/T$,

$$\delta = \pi \nu T + \pi (q \bmod T) + \nu (p \bmod T). \tag{5}$$

From the last equation, it is easy to obtain the following bounds:

$$\pi \nu \leq \left\lceil N/T^2 \right\rceil \tag{6}$$

$$(\pi + 1)(\nu + 1) \geq \left\lfloor N/T^2 \right\rfloor \tag{7}$$

Therefore, $\ell(\pi) + \ell(\nu) \simeq \ell(\pi \nu) \simeq \ell(N/T^2) \simeq 2\alpha - 2(\alpha - c) = 2c$. Because by construction $c$ is a small constant, it is possible to adopt a brute force approach to discover the missing "high-level" coefficients $\pi$ and $\nu$. The brute force search guesses the value of the sum $\pi + \nu$, starting from the lower bound $\left\lfloor \sqrt{2 (\lfloor N/T^2 \rfloor - 1)} \right\rfloor$ (from Equation (7)) and ending at the upper bound $\left\lceil N/T^2 \right\rceil \approx 2^{2c}$ (from Equation (6)).

For any candidate value of the sum $\pi + \nu$, Equation (5) can be transformed by introducing an unknown $x = \pi$, $C = \pi + \nu = x + \nu$, $a = q \bmod T$, $b = p \bmod T$:

$$x (C - x) T + a x + b (C - x) = \delta,$$

that is,

$$T x^2 + (b - a - C T) x + \delta - b C = 0.$$

Because we are looking for integer solutions for $x$ and $C - x$, the brute force attack just tries all values for $C$, in increasing order, and immediately discards any value such that

$$\Delta = (b - a - C T)^2 - 4 T (\delta - b C)$$

is not a square. If the value of $C$ survives, the solutions

$$\left( C T + a - b \pm \sqrt{\Delta} \right) / (2 T)$$

are computed; if either one of the solutions is an integral number, the pair $(x, C - x) = (\pi, \nu)$ is recorded as a candidate solution.

**Example 4** (Continuing Example 3). *By Equation (6), $\pi \nu \leq 1312$, and the search interval for $\pi + \nu$ is $[71, 1312]$. Eventually, the brute force search phase yields the following:*

| $k, \gamma$ | $\delta$ | $(\pi, \nu)$ |
|---|---|---|
| $3, \gamma_1$ | 84596264660542973496163993792600141 64347 | |
| $3, \gamma_2$ | 84575793530685790852756239744558629 31102 | |
| $4, \gamma_1$ | 84600988091501505046247225658251475 43255 | |
| $4, \gamma_2$ | 84555671147368118356972008664461463 61343 | |
| $9, \gamma_1$ | 84600988091501505046247225658251475 43255 | |
| $9, \gamma_2$ | 84562069224052358768979468075414702 24038 | $(48, 26)$ |
| $10, \gamma_1$ | 84601597101429911681771157443238587 42548 | |
| $10, \gamma_2$ | 84546744213875654111562050862224330 61299 | |
| $12, \gamma_1$ | 84601769666084089156400223939926168 56441 | |
| $12, \gamma_2$ | 84544312389746376888876025401865063 13669 | |
| $13, \gamma_1$ | 84595278606813158963027897413995857 33765 | |
| $13, \gamma_2$ | 84588449314558751552140437247309141 33903 | |
| $14, \gamma_1$ | 84586889314736120925968164150962928 63204 | |
| $14, \gamma_2$ | 84594739361504336732556605207808110 38011 | |
| $16, \gamma_1$ | 84586007311455900196018568454500355 87533 | |
| $16, \gamma_2$ | 84585632707459328057429323071963702 72787 | |
| $19, \gamma_1$ | 84589463103559139988234664422619216 95379 | |
| $19, \gamma_2$ | 84598410916078334996540265727910500 78911 | |
| $22, \gamma_1$ | 84600578767623444444784907577201825 48647 | |
| $22, \gamma_2$ | 84561753882414856677461771733050703 00817 | |
| $23, \gamma_1$ | 84601113564314509046368794756800563 75399 | |
| $23, \gamma_2$ | 84563940716907871993092653943096057 04461 | |
| $25, \gamma_1$ | 84592074544273456563827880412508134 32771 | |
| $25, \gamma_2$ | 84577955015438239563020371778819816 37553 | |
| $27, \gamma_1$ | 84599934664237050602988147224150826 25743 | |
| $27, \gamma_2$ | 84573672419298993743469252854270540 04837 | |
| $30, \gamma_1$ | 84584997045859273572924073038919898 99950 | |
| $30, \gamma_2$ | 84593302593938272338189792651421697 41243 | |

*There is only one candidate: $k = 9$, $p \bmod T = 4147809812284244207074134721806864295$, $q \bmod T = 6195194351241176049461146226843224919$, $\pi = 48$, $\nu = 26$.*

### 4.2.3. Recovering the Factors

This phase starts by knowing $N$, $T$, $p \bmod T$, $q \bmod T$, and a list of candidate solutions $(\pi, \nu)$.

For any candidate solution $(\pi, \nu)$, the corresponding

$$p = \pi T + (p \bmod T) \quad \text{and} \quad q = \nu T + (q \bmod T)$$

are computed, then the product $p\,q$ is compared to $N$. One of the candidate solutions certainly yields a factorization of the semi-prime.

**Example 5** (Continuing Example 4). *Finally, we obtain the following:*

$$
\begin{aligned}
p &= \pi T + (p \bmod T) &&= 3138014459056022856355312264049982 4151 \\
q &= \nu T + (q \bmod T) &&= 1739242472351217818232087355161352 44841
\end{aligned}
$$

*and we verify that $p \cdot q = 5457768026042466571066314310612087465251911219452327782472$*
*$1618245793829954991 = N$.*

### 4.3. Analysis

The time complexity of the SSB's recovering procedure can be easily obtained. As explained in the previous subsection, the procedure starts by recovering the "low-level" coefficients by means of an exhaustive search among $O(K)$ possible values for $k$. For every candidate value, the procedure must execute some operations in GF($T$) whose cost is in $O((\log T)^2) = O(\alpha^2)$ and also use the Tonelli–Shanks algorithm to determine if a value $< T$ is a quadratic residue, which costs $O((\log T)^3) = O(\alpha^3)$ [45]. The list of candidate values for $k$ has expected length $K/2$, because in a finite field with an odd number of elements any quadratic residue has two square roots; thus, half of the elements of the field are not the square of another element. Therefore, the "high-level" coefficients recovery phase is executed on $O(K)$ candidate values for $k$ and includes an exhaustive search in an interval of size $O(2^{2c})$; in every iteration the procedure executes a few integer operations on values of bit length $\approx 2(\alpha + c)$; hence, every execution of this phase has a cost in $O(2^{2c} (\alpha + c)^2)$. Finally, the cost of every execution of the third phase is dominated by two multiplications

of values of bit length $\approx \alpha - c$; hence, it is in $O(\alpha^2)$. Summing all up, the worst-case cost of the whole recovering procedure is in $O(K(\alpha + c)^3 2^{2c})$.

The values of the parameters $K$ and $c$ are chosen by the backdoor designer. We would expect that larger values of $K$ and $c$ yield smaller running times for Algorithm 1 and longer running times for the recovery procedure; this intuition is confirmed by the experiments. Anyway, the value of $c$ cannot be made too large or it would be possible to discover the backdoor by just guessing the design key $T$ of bit length $\ell(T) = \alpha - c$. By letting $K \in O(\alpha)$ and $c \in O(\log \alpha)$, for instance, $K \approx \alpha$ and $c = 7$ as suggested in Section 4.1, one obtains a running time for the recovery procedure in $O(\alpha^4)$, that is, a polynomial in the size of the semi-prime.

Experimental Results

In order to confirm that the backdoor works as expected and to assess the execution times with respect to the designer's parameters, the SSB has been implemented in Sage-Math [46] and extensive tests have been performed (the code is open-source and available at https://gitlab.com/cesati/ssb-and-tsb-backdoors.git, accessed on 17 September 2023).

In particular, three values for $\alpha$ have been considered: 512 (the size of factors for RSA-1024), 1024 (RSA-2048), and 2048 (RSA-4096). All tests have been performed by choosing $c = 7$. This means that the escrow keys have sizes 505, 1017, and 2041, respectively. The value of $c$ is so small that detecting the existence of the backdoor by simply guessing the value of the escrow key does not appear to be significantly easier than guessing one of the factors of the corresponding semi-primes. Every test trial involves choosing a value for the parameter $K$, generating an escrow key $T$ and a vulnerable semi-prime, then recovering the factors of the semi-prime by just using the values of the semi-prime and the escrow key. The tests have been executed by varying the parameter $K$ so as to determine a value yielding both fast generations of vulnerable semi-primes and a reasonably quick recovery of the factors.

The tests have been executed on three computational nodes with 16 physical Intel Xeon E5-2620 v4 cores running at 2.1 GHz with 64 GiB of RAM. The nodes are based on the Slackware 14.2 software distribution with a Linux kernel version 5.4.78 and SageMath version 9.1. All tests have properly recovered the factors of the vulnerable semi-primes. Each value of $K \in \{100 \cdot i \mid i = 1, \ldots, 50\}$ has been tested 20 times. The SageMath code is sequential; that is, each test trial runs on a single computation core. Table 1 and Figure 1 report averages and standard deviations of the running times.

**Table 1.** SSB: running times (in seconds, average and standard deviations on 20 trials) for $\alpha = 512, 1024$, and 2048.

| $\alpha = 512$ $K$ | Generation avg. | st.dev. | Recovering avg. | st.dev. | $\alpha = 1024$ $K$ | Generation avg. | st.dev. | Recovering avg. | st.dev. | $\alpha = 2048$ $K$ | Generation avg. | st.dev. | Recovering avg. | st.dev. |
|---|---|---|---|---|---|---|---|---|---|---|---|---|---|---|
| 100 | 4.9 | 4.8 | 2.6 | 1.8 | 100 | 51.0 | 60.3 | 5.4 | 1.9 | 100 | 466.4 | 375.1 | 27.1 | 6.9 |
| 500 | 1.7 | 0.6 | 7.8 | 6.9 | 500 | 17.9 | 7.4 | 18.3 | 13.5 | 500 | 214.0 | 134.7 | 47.4 | 18.6 |
| 1000 | 1.5 | 0.2 | 12.2 | 9.6 | 1000 | 11.6 | 3.2 | 34.9 | 30.7 | 1000 | 151.2 | 61.3 | 71.6 | 37.4 |
| 1500 | 1.5 | 0.1 | 18.0 | 17.1 | 1500 | 11.2 | 1.6 | 33.9 | 37.8 | 1500 | 122.3 | 35.0 | 101.2 | 70.0 |
| 2000 | 1.4 | 0.1 | 10.0 | 9.0 | 2000 | 11.5 | 1.8 | 28.6 | 23.0 | 2000 | 102.7 | 20.3 | 95.8 | 51.7 |
| 2500 | 1.6 | 0.1 | 15.3 | 11.2 | 2500 | 10.6 | 1.3 | 61.2 | 60.0 | 2500 | 112.7 | 25.4 | 113.2 | 84.8 |
| 3000 | 1.7 | 0.1 | 15.7 | 15.7 | 3000 | 11.4 | 1.7 | 45.4 | 63.7 | 3000 | 107.6 | 23.0 | 130.7 | 84.1 |
| 3500 | 1.7 | 0.1 | 26.1 | 30.3 | 3500 | 10.8 | 0.9 | 72.0 | 63.4 | 3500 | 99.5 | 22.6 | 95.5 | 54.8 |
| 4000 | 1.7 | 0.1 | 14.6 | 18.8 | 4000 | 10.7 | 1.5 | 56.0 | 49.0 | 4000 | 90.4 | 9.1 | 143.1 | 104.4 |
| 4500 | 1.6 | 0.2 | 19.0 | 16.6 | 4500 | 10.8 | 1.0 | 42.7 | 41.9 | 4500 | 91.5 | 13.5 | 152.8 | 136.6 |
| 5000 | 1.6 | 0.3 | 24.8 | 19.0 | 5000 | 11.1 | 1.2 | 44.4 | 38.4 | 5000 | 97.3 | 16.9 | 94.5 | 91.5 |

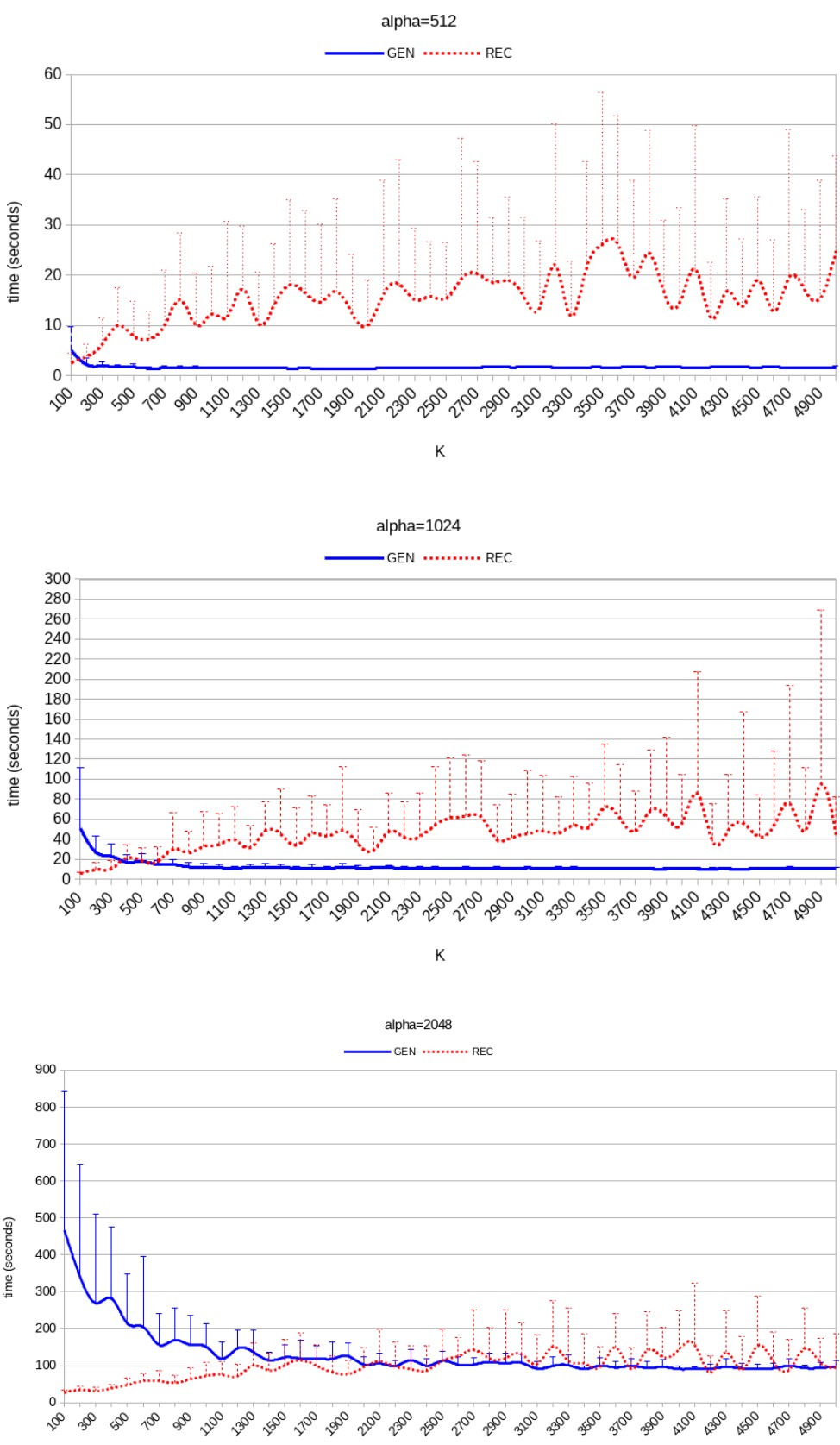

**Figure 1.** SSB: average running times for $\alpha = 512, 1024$, and 2048 (20 trials for each value of *K*). Magnitudes of the standard deviations are shown as vertical bars.

The experimental results confirm that the value of *K* is crucial in determining both the time required to generate a vulnerable semi-prime and the time required to recover the factors. Even if the code has not been optimized at all, the recovery time is reasonably small for all tested values of *K*; hence, the SSB is a practically effective backdoor. However, the generation time is also very important whenever the backdoor mechanism has to be hidden in hardware devices or software programs that are supposed to yield robust, legit semi-primes. While in general larger values of *K* are associated with smaller generation times, there seems to be a threshold value for *K* above which the generation times are essentially constants and near the minimum observed value. From the data shown in Table 1 and Figure 1, *K* may be safely set to values near 500, 1000, and 2000 for $\alpha = 512$, 1024, and 2048, respectively; that is, $K \approx \alpha$.

## 5. TSB: A Backdoor Embedded in a Pair of Semi-Primes

This section describes the TSB (Twin Semi-prime Backdoor), a new proposal for a backdoor embedded in the values of a pair of semi-primes, $N_1$ and $N_2$. These semi-primes are typically to be generated on the same device but can be used independently. For instance, the two semi-primes might be used in two different RSA keys. It is not hard to justify the generation of two different RSA keys. For instance, the user might be told that one RSA key is for business or work usage and the other one is for personal usage. Alternatively, one of the semi-primes can be used to build an RSA key while the other one can be separately stored as an escrow key for the RSA key.

This section first reports how the two semi-primes are generated. Then, it describes the procedure to efficiently factor both semi-primes, provided that the corresponding designer key is known. Finally, the section reports an analysis of the theoretical and practical efficiency of the backdoor.

### 5.1. Generation of the Vulnerable Pair of Semi-Primes

The first step of the generation of a vulnerable pair is choosing a "designer key", which is a secret value that must be known in order to detect and exploit the backdoor. The designer key is a prime *T* of a size slightly smaller than the size of the primes in each semi-prime. Thus, if $\alpha$ is the reference bit length of the primes (e.g., $\alpha = 512$ for RSA-1024), then $\ell(T) \simeq \alpha - c$, where typically $4 \leq c \leq 10$ for $\alpha \leq 2048$; a good value for $\alpha = 512$, 1024, and 2048 appears to be $c = 7$. The backdoor designer must also choose the values of two constants *K* and *B*. The value of *K* is related to the value of $\alpha$, as discussed later; typically, $K \approx \alpha/5$, e.g., $K = 100$, 200, and 400 for $\alpha = 512$, 1024, and 2048, respectively. The constant $B < T$ acts as a detection threshold, so any value for *B* such that $\ell(B) \simeq \alpha - 2c$ is valid.

In order to create a vulnerable pair, four distinct primes, $p_1$, $q_1$, $p_2$, and $q_2$, each of them having a bit length of roughly $\alpha$, must be generated. The backdoor exists whenever the following conditions hold:

**H1.** *There exists a positive integer h with $1 < h \leq K$ such that $q_2 \equiv_T h^2 q_1$.*

**H2.** *There exists a positive integer $k_1$ with $1 < k_1 \leq K$ such that $p_1 \equiv_T h\, k_1\, q_2$.*

**H3.** *There exists a positive integer $k_2$ with $1 < k_2 \leq K$ such that $p_2 \equiv_T k_2\, q_1$.*

**H4.** *The integers $h$, $k_1$, and $k_2$ are all coprimes; that is, $\gcd(h, k_1) = \gcd(h, k_2) = \gcd(k_1, k_2) = 1$.*

**H5.** *$k_2$ is not a divisor of $h\, k_1$ modulo T; that is, $h\, k_1 \not\equiv_T k_2$.*

**H6.** *$(h\, q_1)^2 \bmod T > B$.*

Algorithm 2 can be used to generate the four primes $p_1$, $q_1$, $p_2$, and $q_2$ satisfying the conditions H1–H6 above. Once more, the algorithm is implicitly based on Dirichlet's theorem stating that there are infinitely many primes of the form $a + b\,c$ when $\gcd(a, b) = 1$.

---

**Algorithm 2:** Generation of a vulnerable pair of semi-primes

*GeneratePair*:

> **Input** : $\alpha$, $c$, $K$, $T$
> **Output**: $p_1$, $q_1$, $p_2$, $q_2$
>
> **repeat**
> > generate random primes $q_1$, $p$ of size $\alpha$
> > **for** $h \leftarrow 2$ **to** $K$ **do**
> > > $q_2 \leftarrow p + ((h^2 q_1 - p) \bmod T)$
> > > **if** $q_2$ *is prime* **then**
> > > > break for loop
> > >
> > > **end**
> >
> > **end**
> >
> **until** $q_2$ *is not prime*
> **repeat**
> > $p_1, k_1 \leftarrow GetCorrelPrime(\alpha, q_2, h, T, K, c)$
> >
> **until** $\gcd(k_1, h) \neq 1$
> **repeat**
> > $p_2, k_2 \leftarrow GetCorrelPrime(\alpha, q_1, 1, T, K, c)$
> >
> **until** $\gcd(k_1, k_2) \neq 1$ *or* $\gcd(k_2, h) \neq 1$
> **return** $p_1$, $q_1$, $p_2$, $q_2$

*GetCorrelPrime*:

> **Input** : $\alpha$, $q$, $j$, $T$, $K$, $c$
> **Output**: $p$, $k$
>
> **while** *true* **do**
> > $k \leftarrow$ random value between 2 and $K$
> > $t_1 \leftarrow (k\,j\,q) \bmod T$
> > **for** $p \leftarrow t_1 + 2^{c-3}$ **to** $t_1 + 2^{2\,c-2}$ **do**
> > > **if** $p \equiv_T t_1$ *and* $p$ *is prime* **then**
> > > > **return** $p$, $k$
> > >
> > > **end**
> >
> > **end**
> >
> **end**

---

Finally, the semi-primes are computed as $N_1 = p_1 q_1$ and $N_2 = p_2 q_2$. Observe that $N_1$ and $N_2$ are coprimes, because all factors are necessarily different by construction.

*5.2. Recovering Procedure*

The key idea of the TSB, and also the proof that it works as expected, is its recovering procedure. Formally, the factors of $N_1$ and $N_2$ can be efficiently recovered by knowing in advance only the pair of semi-primes $(N_1, N_2)$ and the designer key $T$. The values of the parameters $\alpha$, $K$, and $c$ may affect the running time of the recovering procedure; however there is no need to know them to recover the factors.

The recovering procedure can be split into four phases:

1.  Recovering "medium-level" coefficients.
2.  Recovering "low-level" coefficients.
3.  Recovering "high-level" coefficients.
4.  Recovering the factors.

Generally speaking, in a practical implementation of the recovering procedure it might be convenient to interleave the executions of these four phases. However, the phases are here described independently to simplify the description of the whole procedure.

**Example 6.** *This is the "running example" for the TSB's recovering procedure. Let $\alpha = 64$, $c = 3$, $K = 100$, and $B = 2^{57}$. Pick as a random secret the 61-bit prime $T = 1350856093440009833$. Then, pick as vulnerable semi-primes $N_1 = 199771249142689629600100193795300988277$ and $N_2 = 330849388672597230630022641974377014199$ (both of bit length 128).*

5.2.1. Recovering "Medium-Level" Coefficients

The recovering procedure starts by assuming to know the following data: $N_1$, $N_2$, and the "secret" prime $T$.

Equations in conditions H1, H2, and H3 enforce the following congruences of $N_1$ and $N_2$ modulo $T$:

$$N_1 \equiv_T p_1 q_1 \equiv_T h k_1 q_2 q_1 \equiv_T h^3 k_1 q_1^2 \tag{8}$$

$$N_2 \equiv_T p_2 q_2 \equiv_T k_2 q_1 q_2 \equiv_T h^2 k_2 q_1^2 \tag{9}$$

It turns out that $N_1$ and $N_2$ are congruent modulo $T$ to two values that have a big common factor, $h^2 q_1^2$. However, the Euclidean algorithm on $N_1 \bmod T$ and $N_2 \bmod T$ does not really help here:

$$\gcd(N_1 \bmod T, N_2 \bmod T) =$$
$$\gcd((h^3 k_1 q_1^2) \bmod T, (h^2 k_2 q_1^2) \bmod T).$$

The point is that the greatest common divisor is relative to the lifted images of the products in the Galois field $GF(T)$, and it is not related to the greatest common divisor of the products $h^3 k_1 q_1^2$ and $h^2 k_2 q_1^2$ in $\mathbb{Z}$.

**Example 7** (Continuing Example 6)**.**

$$\gcd(N_1 \bmod T, N_2 \bmod T) = \gcd(3375980815077368317, 75151731210637471) = 1.$$

To overcome this problem, observe that Equations (8) and (9) also imply the following ones:

$$N_1 \bmod T \equiv_T (h k_1) \cdot \left[ (h^2 q_1^2) \bmod T \right] \tag{10}$$

$$N_2 \bmod T \equiv_T k_2 \cdot \left[ (h^2 q_1^2) \bmod T \right] \tag{11}$$

and therefore there exist two integers $\tilde{k}_1$, $\tilde{k}_2$ such that

$$(N_1 \bmod T) + \tilde{k}_1 \cdot T = (h k_1) \cdot \left[ (h^2 q_1^2) \bmod T \right] \tag{12}$$

$$(N_2 \bmod T) + \tilde{k}_2 \cdot T = k_2 \cdot \left[ (h^2 q_1^2) \bmod T \right] \tag{13}$$

From the last two equations,

$$\gcd((N_1 \bmod T) + \tilde{k}_1 \cdot T, (N_2 \bmod T) + \tilde{k}_2 \cdot T) = \left[ (h^2 q_1^2) \bmod T \right] \tag{14}$$

Observe that dropping $N_1 \bmod T$ from Equation (12) yields

$$\tilde{k}_1 \leq (h k_1) \cdot \frac{(h^2 q_1^2) \bmod T}{T} < K^2.$$

Similarly, from Equation (13),

$$\tilde{k}_2 \le k_2 \cdot \frac{(h^2\,q_1^2) \bmod T}{T} < K.$$

Hence, the sizes of the "medium" coefficients $\tilde{k}_1$ and $\tilde{k}_2$ are so small that they can be quickly recovered by a brute force approach as in Algorithm 3. It is possible to recognize the proper values of $\tilde{k}_1$ and $\tilde{k}_2$ because the size of $\left[(h^2\,q_1^2) \bmod T\right]$ produced by the gcd with the right values is usually much higher than the average value resulting from a gcd with random wrong values. In fact, by condition H6, $(h^2\,q_1^2) \bmod T > B$; hence, the procedure selects any candidate pair of medium-level coefficients $(\tilde{k}_1, \tilde{k}_2)$ for which the greatest common divisor in Equation (14) is between $B$ and $T$. Moreover, the value returned by the Euclidean algorithm with the right values must be a square in the Galois field GF($T$); hence, the procedure may use this condition to filter some false positives. In all test cases, the first value found by this brute force procedure yields a proper factorization result.

---

**Algorithm 3:** Brute force search of the medium-level coefficients

*RecoveryMedCoeff*:

> **Input** : $N_1$, $N_2$, $T$
> **Output**: a list of pairs $(\tilde{k}_1, \tilde{k}_2)$
>
> **for** $s \leftarrow 0$ **to** $\infty$ **do**
> > **for** $\tilde{k}_1 \leftarrow 0$ **to** $s$ **do**
> > > $\tilde{k}_2 \leftarrow s - \tilde{k}_1$
> > > $g \leftarrow \mathtt{gcd}(\tilde{k}_1 \cdot T + N_1 \bmod T, \tilde{k}_2 \cdot T + N_2 \bmod T)$
> > > **if** $B < g < T$ *and* $g \equiv_T \gamma^2$ *for some* $\gamma$ **then**
> > > > add $(\tilde{k}_1, \tilde{k}_2)$ to the list of pairs
> > >
> > > **end**
> >
> > **end**
>
> **end**

---

**Example 8** (Continuing Example 7). *There are only two possible pairs $(\tilde{k}_1, \tilde{k}_2) \in [0, 100^2] \times [0, 100]$ that yield a greatest common divisor higher than $B = 2^{57}$: $(671, 10)$ and $(5277, 79)$. The gcd for the pair $(671, 10)$ is $196865400950880229$, which is the square of $10632559655363908$ modulo T. The gcd for the pair $(5277, 79)$ is $1547721494390890062$: because it is above T, the pair can be discarded.*

5.2.2. Recovering "Low-Level" Coefficients

The previous phase might determine several candidate pairs of medium-level coefficients, and the current phase must be applied to each of them.

This phase starts by assuming to know the following data: $N_1$, $N_2$, $T$, $\tilde{k}_1$, $\tilde{k}_2$, and the value $\gamma^2 = \left[(h^2\,q_1^2) \bmod T\right]$ derived from Equation (14). The value of the "low-level" coefficient $k_2$ can be immediately computed using Equation (13):

$$k_2 = \left((N_2 \bmod T) + \tilde{k}_2 \cdot T\right)/\gamma^2, \tag{15}$$

or, assuming $K < T$, $k_2 = (N_2 \cdot (\gamma^2)^{-1}) \bmod T$ where $\gamma^2 \cdot (\gamma^2)^{-1} \equiv_T 1$.

However, by inverting Equation (12) one obtains the value of the product $h\,k_1$:

$$(h\,k_1) = \left((N_1 \bmod T) + \tilde{k}_1 \cdot T\right)/\gamma^2, \tag{16}$$

or, assuming $K < T$, $(h\,k_1) = (N_1 \cdot (\gamma^2)^{-1}) \bmod T$.

Since both $h$ and $k_1$ are not greater than $K$, their product is below $K^2$. Moreover, by condition H4, $\gcd(h, k_1) = 1$. Because the number of multiplicative partitions of this product does not exceed $K^2$ [47,48], the procedure may exhaustively generate all possible candidate pairs $(h, k_1)$ and apply the forthcoming phases to each of them. When

these phases are performed on the true pair $(h, k_1)$, a proper factorization of $N_1$ and $N_2$ is computed.

**Example 9** (Continuing Example 8). *Two exact integer divisions yield $k_2 = 69$ and $(h\,k_1) = 4606 = 2 \cdot 7^2 \cdot 47$. Therefore, there are six possible pairs $(h, k_1)$, corresponding to the non-trivial subsets of the three values 2, $7^2$, and 47: $(2, 2303)$, $(47, 98)$, $(49, 94)$, $(94, 49)$, $(98, 47)$, and $(2303, 2)$.*

5.2.3. Recovering "High-Level" Coefficients

This phase starts by knowing the following data: $N_1$, $N_2$, $T$, $h$, $k_1$, $k_2$, and $\gamma^2$.

The procedure starts by computing the square root of $\gamma^2 = (h^2\, q_1^2) \bmod T$ in GF($T$); that is, it finds the values whose square is congruent to $\gamma^2$ modulo $T$, typically by means of the Tonelli–Shanks algorithm [43,44]. Because in general any square root has two distinct values in GF($T$), one obtains two possible values $\gamma_1$ and $\gamma_2$ for $(h\,q_1) \bmod T$, where $\gamma_1 \equiv_T T - \gamma_2$. In the following, let $\gamma$ be either $\gamma_1$ or $\gamma_2$; the procedure has to perform this phase with both values and discard the one that yields inconsistent results.

It is now possible to compute the value $q_1 \bmod T$, because $\gamma = (h\,q_1) \bmod T$ means the following:

$$q_1 \bmod T = (\gamma\, h^{-1}) \bmod T \tag{17}$$

where obviously $h^{-1}$ is computed in GF($T$); that is, $h\,h^{-1} \equiv_T 1$.

The value $q_2 \bmod T$ can now be inferred from the equation in condition H1, because

$$q_2 \bmod T = \left( (q_1 \bmod T) \cdot h^2 \right) \bmod T \tag{18}$$

Also, $p_1 \bmod T$ and $p_2 \bmod T$ can be computed from conditions H2 and H3:

$$p_1 \bmod T = (h\,k_1\,(q_2 \bmod T)) \bmod T,$$
$$p_2 \bmod T = (k_2\,(q_1 \bmod T)) \bmod T. \tag{19}$$

**Example 10** (Continuing Example 9). *The square roots of $\gamma^2 = 196865400950880229$ in GF(T) are $\gamma_1 = 10632559655363908$ and $\gamma_2 = 1340223533784645925$. The six possible pairs $(h, k_1)$ and the two possible roots $\gamma_1$ and $\gamma_2$ yield the following 12 cases:*

| $h, k_1, \gamma$ | $q_1 \bmod T$ | $q_2 \bmod T$ | $p_1 \bmod T$ | $p_2 \bmod T$ |
|---|---|---|---|---|
| $2, 2303, \gamma_1$ | 5316279827681954 | 21265119310727816 | 685500817531612520 | 366823308110054826 |
| $2, 2303, \gamma_2$ | 1345539813612327879 | 1329590974129282017 | 665355275908397313 | 984032785329955007 |
| $47, 98, \gamma_1$ | 1264857461085442480 | 499730303802103676 | 1249852184152786057 | 820374834734901808 |
| $47, 98, \gamma_2$ | 85998632354567353 | 851125789637906157 | 101003909287223776 | 530481258705108025 |
| $49, 94, \gamma_1$ | 331038891447662896 | 520995423112831492 | 584496908244388744 | 1227986014848582496 |
| $49, 94, \gamma_2$ | 1019817201992346937 | 829860670327178341 | 766359185195621089 | 122870078591427337 |
| $94, 49, \gamma_1$ | 632428730542721240 | 999460607604207352 | 1148848274865562281 | 410187417367450904 |
| $94, 49, \gamma_2$ | 718427362897288593 | 351395485835802481 | 20200781857444752 | 940668676072558929 |
| $98, 47, \gamma_1$ | 165519445723831448 | 1041990846225662984 | 1168993816488777488 | 613993007424291248 |
| $98, 47, \gamma_2$ | 1185336647716178385 | 308865247214346849 | 181862276951232345 | 736863086015718585 |
| $2303, 2, \gamma_1$ | 466909284818889792 | 171375204382903130 | 454232818686074308 | 1147050503383169489 |
| $2303, 2, \gamma_2$ | 883946808621120041 | 1179480889057106703 | 896623274753935525 | 203805590056840344 |

At this point the procedure knows the values $N_1$, $N_2$, $T$, $q_1 \bmod T$, $q_2 \bmod T$, $p_1 \bmod T$, and $p_2 \bmod T$.

The semi-prime $N_i$ ($i \in \{1, 2\}$) can be written as follows:

$$N_i = p_i\,q_i = (\pi_i\,T + (p_i \bmod T)) \cdot (\nu_i\,T + (q_i \bmod T)), \tag{20}$$

that is, if $\delta_i = (N_i - (p_i \bmod T)\,(q_i \bmod T))/T$,

$$\delta_i = \pi_i\,\nu_i\,T + \pi_i\,(q_i \bmod T) + \nu_i\,(p_i \bmod T). \tag{21}$$

The following bounds can be easily obtained from the last equation:

$$\pi_i \, v_i \le \left\lceil N_i/T^2 \right\rceil \tag{22}$$

$$(\pi_i + 1)\,(v_i + 1) \ge \left\lfloor N_i/T^2 \right\rfloor \tag{23}$$

Therefore, $\ell(\pi_i) + \ell(v_i) \simeq 2\alpha - 2(\alpha - c) = 2c$. Because by construction $c$ is a small constant, the procedure can adopt a brute force approach to discover the missing "high-level" coefficients $\pi_i$ and $v_i$. The brute force search guesses the value of the sum $\pi_i + v_i$, starting from the lower bound $\left\lfloor \sqrt{2(\lfloor N_i/T^2 \rfloor - 1)} \right\rfloor$ (from Equation (23)) and ending at the upper bound $\lceil N_i/T^2 \rceil \approx 2^{2c}$ (from Equation (22)).

For any candidate value of the sum $\pi_i + v_i$, Equation (21) can be transformed by introducing an unknown $x = \pi_i$, $C = \pi_i + v_i = x + v_i$, $a_i = q_i \bmod T$, $b_i = p_i \bmod T$:

$$x\,(C - x)\,T + a_i\,x + b_i\,(C - x) = \delta_i,$$

that is,

$$T\,x^2 + (b_i - a_i - C\,T)\,x + \delta_i - b_i\,C = 0.$$

Because we are looking for integer solutions for $x$ and $C - x$, the brute force attack tries all values for $C$, in increasing order, and immediately discards any value such that

$$\Delta = (b_i - a_i - C\,T)^2 - 4\,T\,(\delta_i - b_i\,C)$$

is not a square. If the value of $C$ survives, the solutions

$$\left( C\,T + a_i - b_i \pm \sqrt{\Delta} \right) / (2\,T)$$

are computed; if either one of the solutions is an integral number, the pair $(x, C - x) = (\pi_i, v_i)$ is recorded as a candidate solution.

**Example 11** (Continuing Example 10). *By Equation* (22), *$\pi_1 \, v_1 \le 110$ and $\pi_2 \, v_2 \le 182$. The search interval for $\pi_1 + v_1$ is $[20, 110]$. The search interval for $\pi_2 + v_2$ is $[26, 182]$. Eventually, the brute force search phase yields the following candidates:*

| $h, k_1, \gamma$ | $c_1$ | $c_2$ | $(\pi_1, v_1)$ | $(\pi_2, v_2)$ |
|---|---|---|---|---|
| $2, 2303, \gamma_1$ | 147882225056116242909 | 244912533420701231951 | | |
| $2, 2303, \gamma_2$ | 147222186060035527550 | 243949765754682004760 | | |
| $47, 98, \gamma_1$ | 146714639142360634749 | 244614821750351042527 | | |
| $47, 98, \gamma_2$ | 147878492694158853453 | 244584070795448038178 | $(9, 12)$ | $(12, 14)$ |
| $49, 94, \gamma_1$ | 147741686848298522541 | 244444700795865219999 | | |
| $49, 94, \gamma_2$ | 147306366554550564348 | 244842826140386624154 | | |
| $94, 49, \gamma_1$ | 147347067872903355989 | 244614821750351042527 | | |
| $94, 49, \gamma_2$ | 147777488784871629677 | 244673613681882690950 | | |
| $98, 47, \gamma_1$ | 147741686848298522541 | 244444700795865219999 | | |
| $98, 47, \gamma_2$ | 147725344017071121644 | 244749828556075164398 | | |
| $2303, 2, \gamma_1$ | 147727922012766098477 | 244772788355361842613 | | |
| $2303, 2, \gamma_2$ | 147298208022831052744 | 244740357969687905399 | | |

*Therefore, there is only one surviving parameter set: $h = 47$, $k_1 = 98$, $k_2 = 69$, $\pi_1 = 9$, $v_1 = 12$, $\pi_2 = 12$, $v_2 = 14$, $p_1 \bmod T = 101003909287223776$, $q_1 \bmod T = 85998632354567353$, $p_2 \bmod T = 530481258705108025$, and $q_2 \bmod T = 851125789637906157$.*

### 5.2.4. Recovering the Factors

This phase starts by knowing $N_i$, $T$, $p_i \bmod T$, $q_i \bmod T$, and a list of candidate solutions $(\pi_i, v_i)$, for $i = 1, 2$. The procedure now works on every semi-prime separately.

For any candidate solution $(\pi_i, v_i)$, it computes the corresponding $p_i = \pi_i\,T + (p_i \bmod T)$ and $q_i = v_i\,T + (q_i \bmod T)$ and then it simply verifies whether $p_i \cdot q_i = N_i$. One of the candidate solutions certainly yields a factorization of the semi-prime.

**Example 12** (Continuing Example 11). *Finally, we obtain the following:*

$$
\begin{aligned}
p_1 &= \pi_1\, T + (p_1 \bmod T) &&= 12258708750247312273 \\
q_1 &= \nu_1\, T + (q_1 \bmod T) &&= 16296271753634685349 \\
p_2 &= \pi_2\, T + (p_2 \bmod T) &&= 16740754379985226021 \\
q_2 &= \nu_2\, T + (q_2 \bmod T) &&= 19763111097798043819
\end{aligned}
$$

*and we verify that*

$$
\begin{aligned}
p_1 \cdot q_1 &= 199771249142689629600100193795300988277 &&= N_1 \\
p_2 \cdot q_2 &= 330849388672597230630022641974377014199 &&= N_2
\end{aligned}
$$

*5.3. Analysis*

The time complexity of the TSB's recovering procedure can be easily obtained. As already explained, the procedure starts by recovering the "medium-level" coefficients by means of an exhaustive search among $K^3$ possible values for the pair $(\tilde{k}_1, \tilde{k}_2)$. For every candidate pair, the procedure must execute the Euclidean algorithm on values of bit lengths up to $\approx \ell(\tilde{k}_1\, T)$, which costs $O(\log(\tilde{k}_1\, T)) = O(\ell(\tilde{k}_1) + \ell(T)) = O(\log K + \alpha - c)$. It may also use the Tonelli–Shanks algorithm to determine if a value $< T$ is a quadratic residue, which costs $O((\log T)^3) = O(\alpha^3)$ [45]. The "low-level" coefficients recovery phase involves a couple of integer divisions on values $\approx \tilde{k}_1\, T$, a factorization of a value $< K^2$, and the generation of up to $K^2$ candidate pairs $(h, k_1)$; hence, the cost of each execution of this recovery phase is $O(\alpha^2 + K^2)$. The "high-level" coefficients recovery phase includes an exhaustive search in an interval of size $O(2^{2c})$; in every iteration the procedure executes a few integer operations on values of bit length $\approx 2(\alpha + c)$; hence, the cost of every execution of this phase is $O(2^{2c}\,(\alpha + c)^2)$. Finally, the cost of every execution of the fourth phase is dominated by four multiplications of values of bit length $\approx \alpha - c$; hence, it is in $O(\alpha^2)$. Summing all up, the worst-case cost of the whole recovering procedure is in $O(K^5\,(\alpha + c)^2\, 2^{2c})$.

The values of the parameters $K$ and $c$ are chosen by the backdoor designer. It is easy to observe that larger values of $K$ and $c$ yield shorter running times for Algorithm 2 and longer running times for the recovery procedure. Anyway, the value of $c$ cannot be made too large, or it would be possible to discover the vulnerability by just guessing the designer key $T$ of bit length $\ell(T) = \alpha - c$. However, experimental results show that larger values of $c$ do not necessarily yield shorter times for the generation phase. By letting $K \approx \alpha/5$ and $c = 7$, as suggested in Section 5.1, one obtains a running time for the recovery procedure in $O(\alpha^7)$, that is, a polynomial in the size of the semi-primes.

Experimental Results

In order to confirm that the backdoor works as expected and to assess the execution times with respect to the designer's parameters, the TSB has been implemented in Sage-Math [46] and extensive tests have been performed (the code is open-source and available at https://gitlab.com/cesati/ssb-and-tsb-backdoors.git, accessed on 17 September 2023).

In particular, three values for $\alpha$ have been considered: 512 (the size of factors for RSA-1024), 1024 (RSA-2048), and 2048 (RSA-4096). All tests have been performed by choosing $c = 7$. This means that the designer keys have sizes 505, 1017, and 2041, respectively. The value of $c$ is so small that detecting the existence of the backdoor by simply guessing the value of the designer key does not appear to be significantly easier than guessing one of the factors of the corresponding semi-primes. Every test trial involves choosing a value for the parameter $K$, generating a designer key $T$ and a pair of vulnerable semi-primes, then recovering the factors of the semi-primes by just using the values of the semi-primes and the designer key. The tests have been executed by varying the parameter $K$ so as to determine a value yielding both fast generations of vulnerable semi-primes and a reasonably quick recovery of the factors.

The tests have been executed on the same computational nodes described in Section 4. All tests have properly recovered the factors of the vulnerable semi-primes. Each value of $K \in \{10 \cdot i \mid i = 1, \ldots, 40\}$ has been tested 20 times. The SageMath code is sequential; that is, each test trial runs on a single computation core. Table 2 and Figure 2 report averages and standard deviations of the running times.

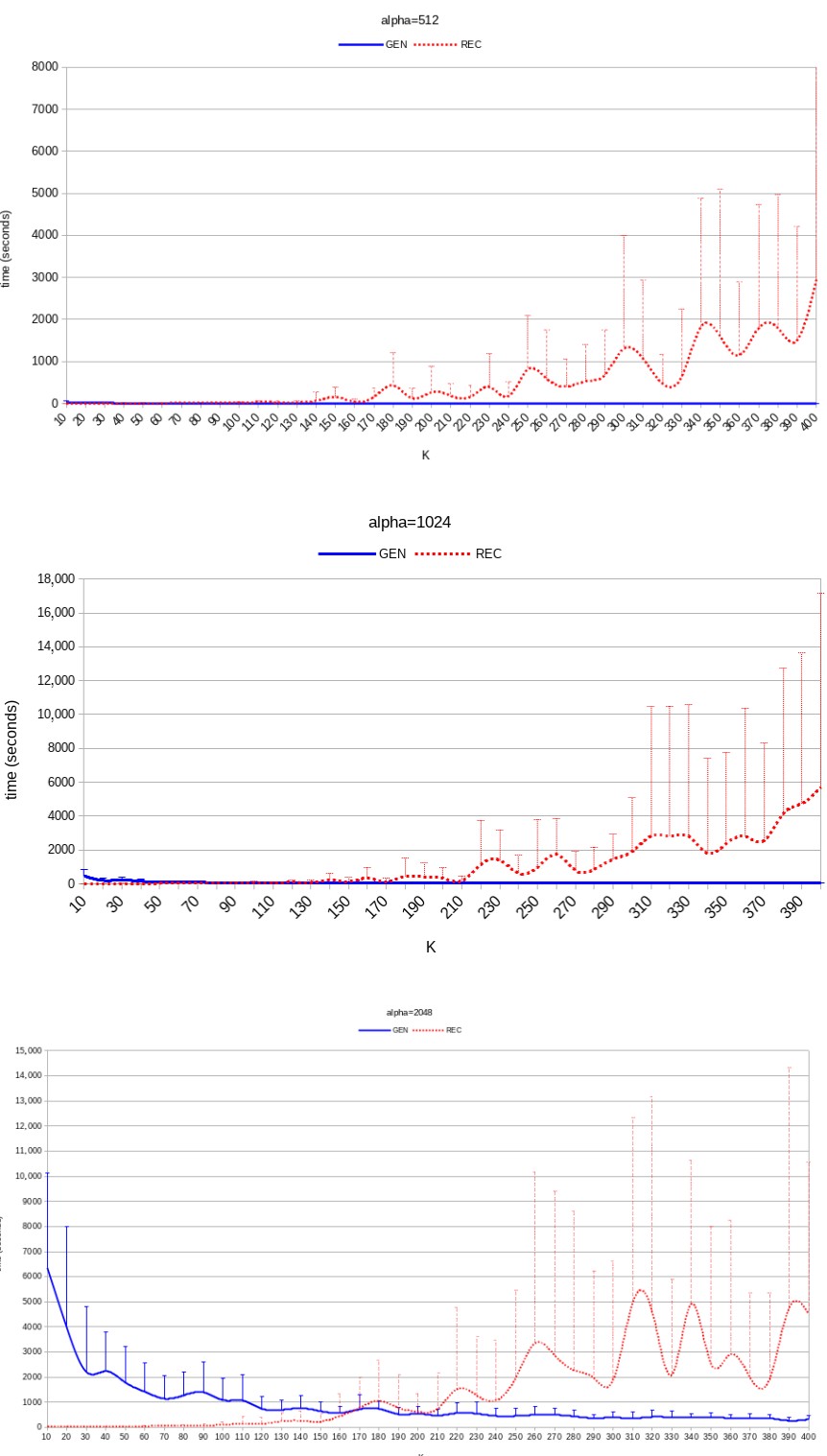

**Figure 2.** TSB: average running times for $\alpha = 512$, 1024, and 2048 (20 trials for each value of $K$). Magnitudes of the standard deviations are shown as vertical bars.

**Table 2.** TSB: running times (in seconds, average and standard deviations on 20 trials) for $\alpha = 512$, 1024, and 2048.

| $\alpha = 512$ K | Generation avg. | st.dev. | Recovering avg. | st.dev. | $\alpha = 1024$ K | Generation avg. | st.dev. | Recovering avg. | st.dev. | $\alpha = 2048$ K | Generation avg. | st.dev. | Recovering avg. | st.dev. |
|---|---|---|---|---|---|---|---|---|---|---|---|---|---|---|
| 10 | 26.6 | 31.8 | 2.7 | 2.7 | 10 | 454.7 | 402.5 | 9.8 | 8.0 | 10 | 6353.2 | 3759.4 | 31.1 | 9.3 |
| 50 | 7.8 | 4.7 | 6.4 | 6.6 | 50 | 95.3 | 69.7 | 23.2 | 20.1 | 50 | 1785.4 | 1429.4 | 43.1 | 12.5 |
| 100 | 6.9 | 2.7 | 23.5 | 24.0 | 100 | 59.3 | 34.4 | 75.1 | 81.3 | 100 | 1086.0 | 887.3 | 104.4 | 108.7 |
| 150 | 5 | 1.5 | 151.9 | 237.0 | 150 | 57.8 | 35.2 | 133.2 | 286.4 | 150 | 647.1 | 376.5 | 236.3 | 290.4 |
| 200 | 5.6 | 2.6 | 270.8 | 612.9 | 200 | 43.5 | 23.9 | 315.6 | 641.3 | 200 | 544.3 | 277.6 | 619.9 | 729.6 |
| 250 | 3.9 | 1.3 | 811.9 | 1277.7 | 250 | 45.8 | 21.8 | 981.4 | 2810.1 | 250 | 456.8 | 305.3 | 1976.0 | 3493.5 |
| 300 | 3.9 | 1.3 | 1309.1 | 2681.1 | 300 | 38.2 | 16.3 | 1905.0 | 3161.7 | 300 | 395.4 | 236.6 | 1910.5 | 4716.5 |
| 350 | 4.2 | 1.2 | 1598.7 | 3494.7 | 350 | 38.1 | 26.1 | 2388.7 | 5378.7 | 350 | 407.6 | 155.3 | 2537.8 | 5460.6 |
| 400 | 4.5 | 2.3 | 2946.3 | 5371.8 | 400 | 35.9 | 12.7 | 5695.4 | 11,449.7 | 400 | 321.1 | 140.0 | 4541.4 | 6038.2 |

The value of $K$ is crucial in determining both the time required to generate a pair of semi-primes and the time required to recover the factors. The experimental results show that, even if the SageMath code is not optimzed, the recovery time is reasonably small for all tested values of $K$; hence, the TSB is a practically effective backdoor. However, generation time is also very important whenever the backdoor mechanism has to be hidden in hardware devices or software programs that are supposed to yield robust, legit semi-primes. While in general larger values of $K$ are associated with smaller generation times, there seems to be a threshold value for $K$ above which the generation times are essentially constants and near the minimum observed value. From the data shown in Table 2 and Figure 2, $K$ can be safely set to values near 100, 200, and 400 for $\alpha = 512$, 1024, and 2048, respectively; that is, $K \approx \alpha/5$.

## 6. Conclusions

I presented a new idea for designing backdoors in cryptographic systems based on the integer factorization problem. The idea consists of introducing some mathematical relations among the factors of the semi-primes based on a large prime chosen by the designer. A first algorithm, the SSB, can be used to implement a symmetric backdoor; hence, the designer key acts as a pure escrow key that must be kept hidden from the owner of the generated keys (in order to hide the vulnerability) and from third-party attackers. Another proposed algorithm, the TSB, injects a vulnerability in a pair of distinct semi-primes and may be used to implement either a symmetric backdoor or an asymmetric backdoor. Implementations of both the SSB and TSB in SageMath allowed me to conduct extensive experiments to determine optimal values for the trade-off between the generation time of the vulnerable semi-primes and the recovery time when exploiting the backdoors. The SageMath code has not been optimized; however, even for large RSA-4096 keys the recovery time is reasonably small (a few hours, at worst, on a single computation core).

*Future Works*

It does not seem to be hard to plug an asymmetric cipher into both the SSB and TSB, as performed for the Anderson's backdoor by Markelova [19]; this may be a future evolution of the present work. Moreover, a crucial point is minimizing the generation time of the vulnerable semi-primes. The generation algorithms presented here are not very sophisticated or optimized, because basically they generate random values in the hope to find the proper primes satisfying the mathematical conditions of the backdoors. It would be desirable to obtain generation times similar to those of legit public key generators. However, an analysis of the performances of semi-prime generators likely depends on the characteristics of the underlining pseudo-random number generators, which also may depend on external factors, such as the amount of entropy collected by the system (see, for example, Linux's PRNG). Such analysis is not simple; hence, it has to be deferred to a future work.

**Funding:** This research received no external funding.

**Data Availability Statement:** The software developed for this research is open source and available at https://gitlab.com/cesati/ssb-and-tsb-backdoors.git (accessed on 17 September 2023).

**Acknowledgments:** I gratefully thank Epigenesys s.r.l. for providing the computational resources used in the experimental evaluation of the algorithms.

**Conflicts of Interest:** The author declares no conflict of interest.

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
