# Peer review of "A New Idea for RSA Backdoors"

_cryptography, doi:10.3390/cryptography7030045_

Round 1

Reviewer 1 Report

1. The introduction part should be structured by referring to  

2. The caption of any table should be placed above it.

3. The pseudo-code should be presented with environment algorithm NOT figure.

https://www.ctan.org/pkg/algorithms

4. The conclusion part should be written in ONE paragraph by referring to 

http://writingcenter.unc.edu/handouts/conclusions

5. Some concrete examples, including the intermediate data, should be presented to assist the readers to understand the key contribution of the paper.

NA

Reviewer 2 Report

It is an excellent paper. Quantum computing using a variation on Grover's algorithm or variation on Shor's algorithm, or another quantum computing approach may render this technique moot for building secure backdoors. That is not likely to occur before 2030 based on current estimates. This paper was originally published as an arXiv preprint which does not appear in the references

Cesati, Marco. "A new idea for RSA backdoors." arXiv preprint arXiv:2201.13153 (2022).

Reviewer 3 Report

Backdoors in RSA consist of strategies for a key generator to build RSA parameters enough strong for any attackers but weak for the proper key generator. A new method to inject backdoors in RSA are introduced by exploiting mathematical congruences among the factors of the RSA modulus and a large prime number, acting as a "designer key" or "escrow key". As usual, the backdoors are built by considering just one RSA modulus or a sequence of RSA modulus through a fixed designer key. The author considers both cases. An appreciable contribution is the experimentation platform, made with SageMath, in which the author has validated his constructions. The author tests the robustness of his constructions against well known Anderson's backdoor construction attacks. Since the backdoors may involve random searches for appropriate multipliers, the methods may involve lengthy calculations. The author poses the minimizations of these search procedure as a forthcoming research.

In particular, two different backdoors are proposed, one targeting a single semi-prime and the other a pair of semi-primes. Both methods contain original contributions to the commonly known backdoor methods to break RSA and the provided set of programs to validate the proposed methods is, as well, an appreciable contribution.

Author Response

Thank you for your time and your comments: I have really appreciated them.